# Role of MRI-Derived Radiomics Features in Determining Degree of Tumor Differentiation of Hepatocellular Carcinoma

**DOI:** 10.3390/diagnostics12102386

**Published:** 2022-09-30

**Authors:** Sanaz Ameli, Bharath Ambale Venkatesh, Mohammadreza Shaghaghi, Maryam Ghadimi, Bita Hazhirkarzar, Roya Rezvani Habibabadi, Mounes Aliyari Ghasabeh, Pegah Khoshpouri, Ankur Pandey, Pallavi Pandey, Li Pan, Robert Grimm, Ihab R. Kamel

**Affiliations:** 1Department of Radiology, University of Arkansas for Medical Sciences, 4301 W. Markham St., Little Rock, AR 72205, USA; 2Department of Radiology, Johns Hopkins Hospital, 600 N Wolfe St., Baltimore, MD 21287, USA; 3Department of Radiology, University of Florida College of Medicine, 1600 SW Archer Rd., Gainesville, FL 32610, USA; 4Department of Radiology, Saint Louis University, 1201 S Grand Blvd, St. Louis, MO 63104, USA; 5Department of Radiology, University of Washington Main Hospital, 1959 NE Pacific St., 2nd Floor, Seattle, WA 98195, USA; 6Department of Radiology, University of Maryland Medical Center, 22 S Greene St., Baltimore, MD 21201, USA

**Keywords:** carcinoma, hepatocellular, machine learning, neoplasm grading, diffusion magnetic resonance imaging, contrast media

## Abstract

***Background***: To investigate radiomics ability in predicting hepatocellular carcinoma histological degree of differentiation by using volumetric MR imaging parameters. ***Methods***: Volumetric venous enhancement and apparent diffusion coefficient were calculated on baseline MRI of 171 lesions. Ninety-five radiomics features were extracted, then random forest classification identified the performance of the texture features in classifying tumor degree of differentiation based on their histopathological features. The Gini index was used for split criterion, and the random forest was optimized to have a minimum of nine participants per leaf node. Predictor importance was estimated based on the minimal depth of the maximal subtree. ***Results***: Out of 95 radiomics features, four top performers were apparent diffusion coefficient (ADC) features. The mean ADC and venous enhancement map alone had an overall error rate of 39.8%. The error decreased to 32.8% with the addition of the radiomics features in the multi-class model. The area under the receiver-operator curve (AUC) improved from 75.2% to 83.2% with the addition of the radiomics features for distinguishing well- from moderately/poorly differentiated HCCs in the multi-class model. ***Conclusions***: The addition of radiomics-based texture analysis improved classification over that of ADC or venous enhancement values alone. Radiomics help us move closer to non-invasive histologic tumor grading of HCC.

## 1. Introduction

Hepatocellular carcinoma (HCC) is among the most common causes of cancer in the world, and with an increase in its incidence, it now became the second most common cause of cancer-related mortality worldwide [1]. Despite the new surgical and chemotherapeutic techniques in treating HCC tumors, treatment outcome is still suboptimal. Several studies have shown that HCC tumors’ histological grade is one of the critical factors in determining treatment outcome and patients’ overall survival [2]. For this reason, determining the degree of differentiation in HCC tumors at presentation can help practitioners to make an optimal treatment strategy and predict the outcomes more accurately [3]. The characteristic features of HCC on contrast-enhanced MRI are the gold standard in HCC diagnosis [4]. Lesions are typically hypervascular in the hepatic arterial phase, with washout and rim enhancement in the venous/delayed phases [4]. Liver-specific contrast agents like gadoxetate disodium (Gd-EOB-DTPA) can be added to conventional MR imaging, which helps to better visualize liver vasculature and assess data regarding hepatocytes function. These features of Gd-EOB-DTPA have resulted in improved accuracy in HCC diagnosis, as compared to other MRI techniques and dynamic contrast-enhanced computed tomography [5]. Additionally, new advancements in imaging technology have provided opportunities to investigate the microenvironment of tumors. Methods like dynamic contrast-enhanced (DCE) MRI and diffusion-weighted imaging (DWI) facilitated the accurate examination of tumors’ metabolism and proliferation [6]. DWI can provide information regarding tissue cellularity, necrosis, and cell membrane integrity [7]. On the other hand, Gadoxetic acid is a liver-specific contrast agent that has been regularly used in liver MRI scans and emerged as an important tool for HCC diagnosis [8]. Published data regarding the role of DWI and DCE-MRI in prediction of tumor grade and/or microvascular environment in HCC were inconsistent. In addition, while mean apparent diffusion coefficient (ADC) values are typically used, others have suggested the use of minimum ADC or true diffusion as possible measures. Similarly, in the use of DCE-MRI, both quantitative and semi-quantitative parameters have been explored [9,10,11,12]. New developments in the field of artificial intelligence (AI) and machine learning have made it possible to use analytical algorithms to extract a large number of features from imaging data [13]. The use of radiomics may help build more accurate and reproducible results and additionally has the potential to provide new information regarding tumors’ texture and other characteristics [14]. Radiomics has been performed in a variety of imaging sequences for tumor differentiation and grading, including T1-weighted images, T2-weighted images, ADC maps, DCE maps, and T1 maps [15,16,17,18]. In this study, we aimed to use radiomics on imaging data from volumetric ADC and volumetric venous enhancement maps to predict the histopathologic degree of differentiation in HCC tumors.

## 2. Materials and Methods

### 2.1. Study Population

This was a monocentric, retrospective study compliant with Health Insurance Portability and Accountability Act (HIPAA) policies. Informed patient consent was waived by our Institutional Review Board (IRB). Inclusion criteria are shown in Figure 1. A total of 129 HCC patients with baseline MRI and pathologic report were identified between January 2003 and June 2017. All lesions were hypervascular on hepatic arterial phase imaging, with washout on portal venous/delayed phases and rim enhancement.

### 2.2. Histopathology

All sample tissues were obtained following transplantation, resection, or biopsy of the liver and stained with Hematoxylin-eosin. A pathology resident evaluated all tissue samples, and the findings were confirmed by an attending pathologist. Both pathologists were blinded to the MRI findings. Tumor differentiation was defined as well, moderately, or poorly differentiated based on Edmondson–Steiner’s system [19] and using the most dominant grading among the specimens.

### 2.3. MRI Technique and Tumor Segmentation

Our standard protocol was used to perform MR imaging, as shown in Table 1.

Tumor segmentation was performed as reported in prior studies (Appendix A).

Up to 4 dominant lesions in each patient were segmented using semi-automated “Random-Walker” 3D s algorithm on portal venous phase (PVP) images by a postdoctoral fellow (---) with >2 years of experience in using the software, who was blinded to pathology results. Then target tumor, histograms, and volumetric statistics were obtained for volumetric ADC and VE (vADC and vVE) parameters [20,21].

### 2.4. Feature Extraction

An in-house-developed MATLAB-based (R2017b, Mathworks Inc., Natick, MA, USA) program was used to perform analysis using the texture analysis toolbox (https://github.com/mvallieres/radiomics, accessed on 12 August 2021), which extracted 95 texture features from the segmented tumor (Figure 2).

The analysis included 9 global features, in addition to 43 features each for ADC and VE maps: 3 histogram-based features, 9 gray-level co-occurrence matrix features (GLCM), 13 gray-level run-length matrix features (GLRLM), 13 gray-level size-zone matrix (GLSZM) features, and 5 neighborhood gray-tone difference matrix (NGTDM) features.

Global features included the mean, maximum, and minimum voxel intensity for both ADC and VE maps. In addition to the 6 global features above, ADC-map derived tumor solidity, surface area, and volume were also included. The texture analysis consisted of the following histogram-based features: Variance, skewness, and kurtosis. Histogram-based features, unlike all other texture features, were spatially invariant such that the arrangement of the pixels relative to one another did not affect the analysis.

The GLCM, GLRLM, GLSZM, and NGTDM matrix features included are shown in Table 2.

### 2.5. Grey Level Discretization

To investigate the dependence of texture features on the number of gray levels (Ng), texture features were extracted with resampled Ng values of 16, 32, 64, 128, and 256. Of these, 64 was the optimal value in our analysis and provided the best classification performance.

### 2.6. Statistical Analysis

In our analysis, we compare the ability of radiomics to classify tumor differentiation compared to the mean ADC and venous enhancement values alone. Once the features were extracted using texture analysis, statistical machine learning was used to identify the performance of the combined set of texture features in the classification of tumors into well, moderately, and poorly differentiated classes. We used both a two-class (well vs. moderate/poor) and a multi-class (well vs. moderate vs. poor) random forest classification algorithm [22] for that purpose and then tuned the algorithm to optimize the parameters. The Gini index was used for split criterion. Parameters used in the multi-class classification were: Minimum leaf node size = 9, number of variables tried at each split = 44, and number of decision trees = 500. Predictor importance was estimated based on the minimal depth of the maximal subtree. If a predictor is influential in prediction, then the variable is likely to occur nearer to the root rather than the leaf nodes. The out-of-bag error from the random forest algorithm was used as the metric to quantify classification error and assess performance (Figure 3).

OOB estimates measure the prediction error of random forest models utilizing bootstrap aggregating (bagging). Bagging uses subsampling with replacement to create training samples for the model to learn from. Bootstrap aggregating allows one to define an out-of-bag estimate of the prediction performance improvement by evaluating predictions on those observations which were not used in the building of the base learner. The ability of the model to distinguish well from moderate/poorly differentiated tumors was quantified using the area under the receiver operating characteristic (ROC) curve (AUC) and misclassification error rates. OOB AUC’s between the models were compared to assess if the differences in AUC were significant (pROC package). A chi-squared test was used to compare if the differences in the OOB misclassification rates were significant. 

Stata software (Version 15, StataCorp, College Station, TX, USA) was used for the analysis of demographic and clinical parameters. Normality of these variables was assessed by Shapiro–Wilk test, and appropriate statistical tests were used for univariate associations. Kruskal–Wallis and Chi square tests were used for continuous and categorical parameters, respectively. All *p*-values were considered statistically significant at *p* < 0.05.

## 3. Results

### 3.1. Descriptive Findings

A total of 171 HCC lesions were assessed in 129 patients. The median time between MRI and biopsy was 30 days (range 12–68 days). The median time between MRI and resection/transplantation was 191 days (range 93–316 days). All demographic characteristics (Age, Sex, Race) were similar between well, moderate, and poorly differentiated groups. There was no significant difference in patients’ Child scores between the three groups. There was no correlation between lesion location and its degree of differentiation. Alpha-Fetoprotein (AFP) was significantly higher in poorly differentiated HCCs as compared to well and moderately differentiated HCCs (*p* = 0.003). Table 3 summarizes the demographic and clinical characteristics of all patients.

### 3.2. Radiomics Feature Extraction

Of the total of 95 texture features that were extracted, ADC features performed better in distinguishing well differentiated from poorly differentiated HCCs. The top radiomics features estimated from the minimum depth of the maximal subtree for the multi-class random forest classifier were also extracted (Figure 4. Top features as obtained from the multi-class classification algorithm were as follows: (1) Mean ADC value, (2) ADC gray-level zone-length matrix low gray level zone emphasis, (3) enhancement NGTDM coarseness, (4) ADC global variance, and (5) ADC gray-level zone-length matrix short zone low gray level emphasis.

### 3.3. Classification Ability of Radiomics-Based Model

The mean ADC and venous enhancement map alone had an out-of-bag error rate of 39.8%, the error decreased to 32.8% (*p* < 0.01) with the addition of the radiomics features in the multi-class model.

The AUC improved significantly from 75.2% to 83.2% (*p* = 0.03) with the addition of the radiomics features for distinguishing well differentiated from moderate/poorly differentiated HCCs in the multi-class model (Figure 5).

In the two-class problem, the addition of radiomics features decreased the overall error rate from 27.5% to 22.2% (*p* < 0.01) and increased the AUC from 77.9% to 81.5% (*p* = 0.18). (Figure 6).

## 4. Discussion

In this study, we used a radiomics framework that included texture extraction followed by statistical machine learning to analyze the role of volumetric ADC and venous enhancement maps in determining the histopathology of HCC. Using a random forest classification algorithm, we demonstrated that ADC radiomics features were among the top classifiers in variable importance ranking for classifying HCC tumors as compared to VE features. From all the features, we identified five that were superior to other features in tumor classification. The addition of radiomics textures improved classification compared to ADC or venous enhancement values alone, indicating a potential role for radiomics in non-invasive histologic grading of HCC tumors.

To date, despite the major improvement in surgical and chemotherapy techniques, survival of HCC patients remains poor. High recurrence rate is among the main reasons for poor survival in treated patients [23]. Poorly differentiated HCCs have been reported to have higher recurrence rates as compared to well and moderately differentiated tumors and with worse overall survival [2]. Poorly differentiated HCCs need more extensive resection in order to reduce the recurrence rate after surgery [24]. Additionally, more frequent follow-ups might be needed in poorly differentiated HCCs for early detection of recurrence and metastasis [25]. Fine-needle aspiration (FNA) biopsy is the pre-operative gold standard in determining histopathological grading of HCC. However, it is an expensive and invasive method with an increased risk of adverse outcomes, including bleeding, tumor seeding, and sampling errors [26]. Due to the heterogeneous nature of HCC, tissue samples provided by biopsy might not be a good representative of the entire tumor. Therefore, identifying a non-invasive and accurate method to assess tissue characteristics of tumors can be of great importance.

The radiomics’ potential role has been studied for predicting treatment response, recurrence, and overall survival in HCC patients [27,28]. Additionally, several studies have exploited radiomics on pre-contrast T1-W, post-contrast T1-W, T2W MRI, and also contrast-enhanced computed tomography (CE-CT) imaging to distinguish between well differentiated and moderately/poorly differentiated HCC [18,29,30]. Zhang et al. [31] used DWI radiomics features in combination with T1- and T2-weighted imaging to predict microvascular invasion (MVI) in HCC. Their results showed that radiomics features can classify MVI HCCs and non-MVI HCCs with an accuracy of 78% in the training cohort and 82% in the validation cohort.

The results of a study by Hectors et al. showed no association between ADC radiomics features and degree of tumor differentiation of HCCs [32]. In their study, they placed an ROI on the HCC tumor on a single slice of the ADC map. Therefore, the results might not be representative of the whole tumor.

Another study done by Zhou et al. used a convolutional neural network to extract deep features from log maps resulting from three-b-value images of DWI. Their results showed higher performance of their model for HCC grading as compared to ADC maps [26]. Gadolinium-based contrast agents have been used previously in differentiating HCC from benign lesions [33], and studies showed inconclusive results regarding their role in distinguishing HCC histopathological grading [34,35]. With the use of machine learning, several studies exploited contrast-enhanced imaging in predicting early and late recurrence in HCC patients [36,37].

We exploited random forest (RF) classifiers in our study as the machine learning algorithm of choice. RFs have several inherent advantages over other classifiers like statistical logistic regression techniques that are routinely used: (1) They deal well with non-linear associations as the tree method identifies several cut-points during branching, (2) the variables do not have to be normally distributed, (3) the algorithm provides added robustness to prevent overfitting by randomly choosing a subset of variables at each node and also choosing a subset of patient data for each tree, and (4) the algorithm also provides the importance of each variable used.

There were some limitations to our study. First, this study was retrospective, and patients’ data were recorded for several years. The variation in study setting over time is an inherent limitation to all retrospective studies. This limitation was minimized by adhering to consistent protocols in our institute. The other limitation is the use of either biopsy or the entire tumor examination for histopathology grading. This could have potentially affected our findings as the accuracy of biopsy might be lower than tumor excision. However, we performed subgroup analysis in these two groups and also adjusted our final multivariable models for the sampling method, which demonstrated consistent results. Future prospective studies with a better control of confounders can determine other predictors of tumor grading and patients’ survival.

In conclusion, the addition of radiomics-based texture analysis improved classification over and above that of ADC or venous enhancement values alone. Radiomics, by better capturing the tumor microenvironment, may assist in non-invasive histologic grading of HCC tumors.

## Figures and Tables

**Figure 1 diagnostics-12-02386-f001:**
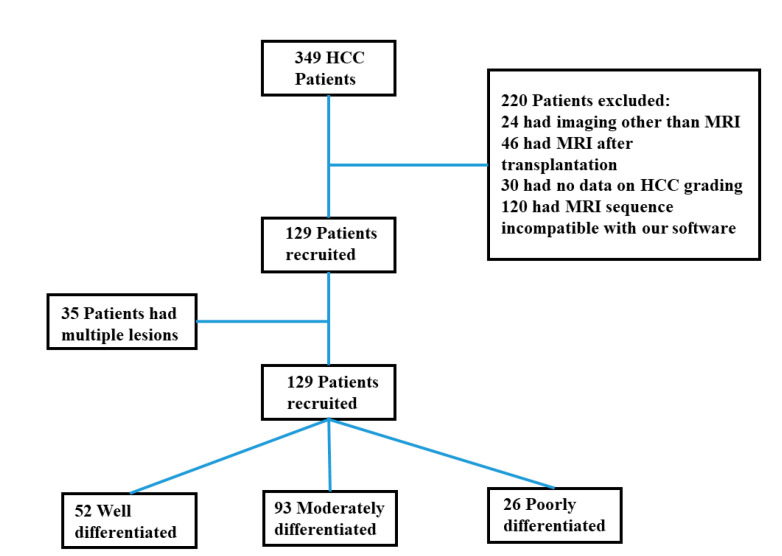
Flowchart shows initial study population, patients excluded, and final study population.

**Figure 2 diagnostics-12-02386-f002:**
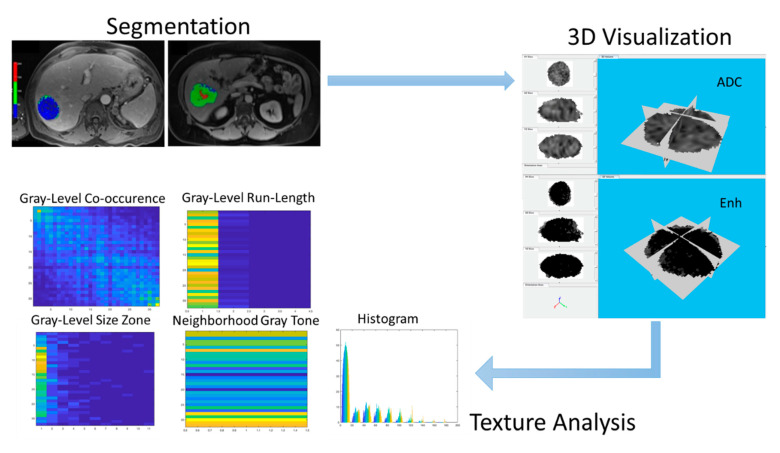
The overall schematic showing tumor radiomics from segmentation to 3D visualization to radiomics on gray-level 3D tumors.

**Figure 3 diagnostics-12-02386-f003:**
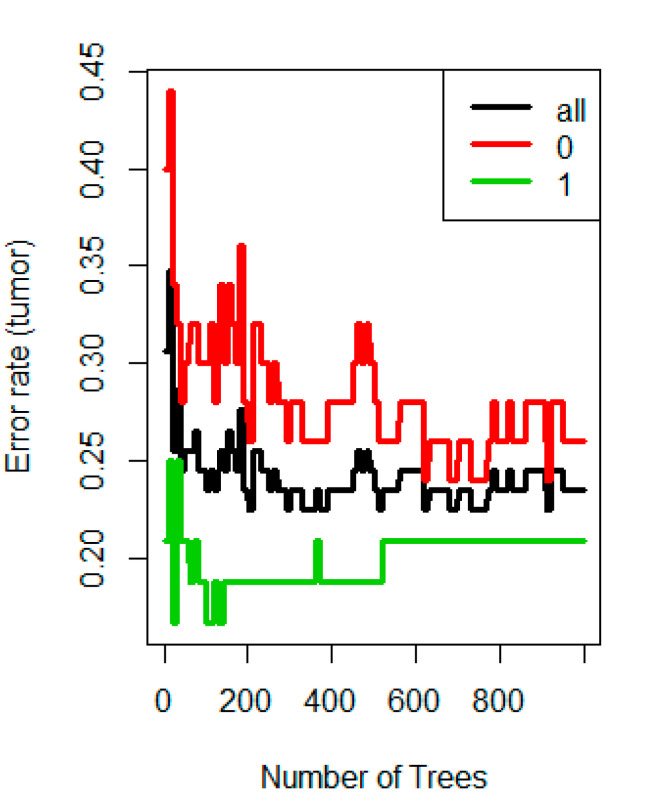
The effect of increasing the number of trees used for building the forest. The out-of-bag error rate stabilizes after around 400 trees in our study.

**Figure 4 diagnostics-12-02386-f004:**
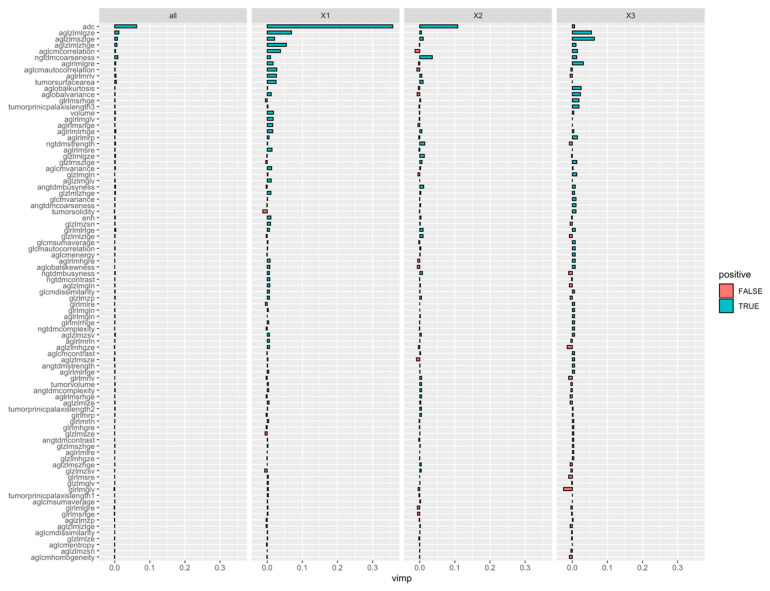
A variable importance plot showing that while mean ADC effectively differentiates well (X1) from moderate (X2) and poor (X3), the addition of radiomics features is particularly helpful in improved differentiation between moderate and poor categories.

**Figure 5 diagnostics-12-02386-f005:**
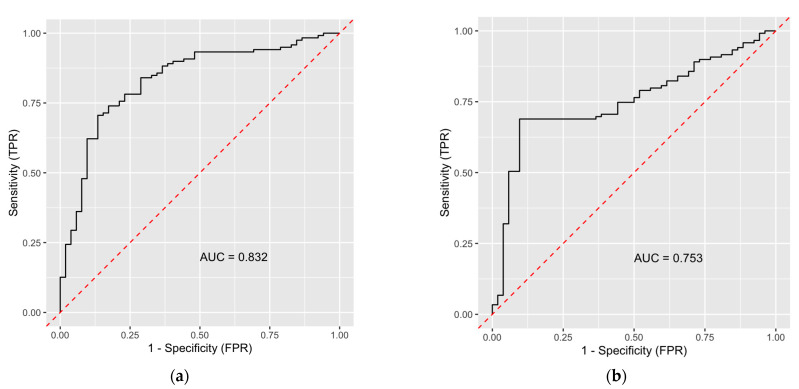
ROC curves showing the performance of models for classifying well from moderate/poorly differentiated tumors with (**a**) and without (**b**) radiomics-derived texture features in the multi-class model.

**Figure 6 diagnostics-12-02386-f006:**
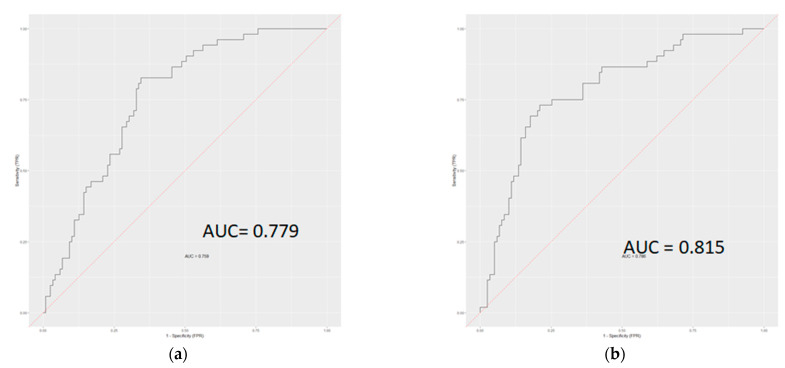
ROC curves showing the performance of models for classifying well from moderate/poorly differentiated tumors without (**a**) and with (**b**) radiomics-derived texture features in the two-class model.

**Table 1 diagnostics-12-02386-t001:** MR scanning protocol parameters.

Parameters	T1WI	T2WI	DWI	DKI	T2*WI
Sequence	FSE	FSE	SS-EPI	SS-EPI	Multiecho GRE
Orientation	Oblique axial	Oblique axial, sagittal and coronal	Oblique axial	Oblique axial	Oblique axial
Repetition time (msec)	500	5629	4000	3000	100
Echo time (msec)	75	85	75	75	2.7, 6.8, 10.9, 15.1, 19.2, 23.3, 27.4, 31.5, 35.6, 39.7, 43.8, 48.0, 52.1
FOV (mm^2^)	380 × 380	200 × 200	400 × 400	360 × 252	240 × 192
Matrix (mm^2^)	320 × 224	448 × 314	160 × 128	128 × 128	192 × 160
Slice Thickness (mm)	5	3	3	3	3
Slice Gap (mm)	1	0	0	0	0
NEX	2	4	12	2	1
b-value (s/mm^2^)	N/A	N/A	0, 800	0, 1000, 2000	N/A
Bandwidth (kHz)	62.50	31.3	250	250	31.3
Scan time	1 min 44 s	4 min 4 s	2 min 32 s	5 min 9 s	1 min 22 s

GRE = gradient-recalled echo; SS-EPI = single-shot echo planar imaging; FSE = Fast spin-echo; FOV = field-of-view; NEX = number of excitations.

**Table 2 diagnostics-12-02386-t002:** Extracted Texture Features from the Tumor Segments.

Texture Feature Category	Extracted Features in Each Category
Global Features	Mean, Maximum, and Minimum (for both ADC and VE), Tumor Solidity, Surface Area, and Volume
Histogram-based Features	Variance, Skewness, and Kurtosis
Gray Level Co-occurrence Matrix Features (GLCM) *	Contrast, Correlation, Energy, Variance, Sum average, Dissimilarity, Autocorrelation, Entropy, and Homogeneity
Gray Level Run Length Matrix Features (GLRLM) **	Short-run emphasis (SRE), Long-run Emphasis (LRE), Gray-level non-uniformity (GLN), Run-length non-uniformity (RLN), Run Percentage (RP), Low Gray-level Run Emphasis (LGRE), High Gray-level Run Emphasis (HGRE), Short Run Low Gray-level Emphasis (SRLGE), Short Run High Gray-level Emphasis (SRHGE), Long Run Low Gray-level Emphasis (LRLGE), Long Run High Gray-level Emphasis (LRHGE), Gray-level Variance (GLV), and Run Length Variance (RLV)
Gray Level Size Zone Matrix Features (GLSZM) ***	Small Zone Emphasis (SZE), Large Zone Emphasis (LZE), Gray-level non-uniformity (GLN), Zone Size non-uniformity (ZSN), Zone percentage (ZP), Low Gray-level Zone Emphasis (LGZE), High Gray-level Zone Emphasis (HGZE), Small Zone Low Gray-level Emphasis (SZHGE), Large Zone Low Gray-level Emphasis (LZLGE), Large Zone High Gray-level Emphasis (LZHGE), Gray-level Variance (GLV), and Zone Size Variance (RLV)
Neighborhood Gray-tone Difference Matrix (NGTDM) ****	Mean, Variance, Kurtosis, Strength, and Skewness
	* GLCM elements in row (i) and column (j) represent the frequency in which a given gray level of value (i) is horizontally adjacent to gray-level (j). For the purposes of this study, these calculations were performed in vertical, horizontal, 45°, and 135° directions, which were then averaged together to minimize directional dependence in the samples. ** Rows (i) represent the gray-levels while the columns (j) represent the run-length, or the consecutive number of pixels with a particular gray-level. Elements within the matrix represent the frequency of pixel line segments with a run-length (j) and a gray-level (i). *** Rows (i) represent the gray-levels while the columns (j) represent the 3D zone-size, or the consecutive number of 3D zones with a particular gray-level. Elements within the matrix represent the frequency of zones with a zone-size (j) and a gray-level (i). **** these features provide a histogram of the absolute gradient values in the tissue segment. In this analysis, differences in all pixel values within a tumor segment were analyzed using a 3 × 3 neighborhood

GLCM features were calculated based on a symmetric matrix with rows (i) and columns (j) ranging from 0 to Ng, such that Ng is equal to the number of gray-levels within the image.

**Table 3 diagnostics-12-02386-t003:** Demographic and clinical parameters in all HCC patients, and in subgroups of well, moderately, and poorly differentiated histopathology.

Parameter		Total	Degree of Differentiation			*p* Value ^Δ^
			Well	Moderate	Poor	
Age *, years		62 (57–68)	61 (57–69)	62 (58–66)	63 (57–73)	0.68
Sex, n (%)	Male	102 (79%)	28 (21.7%)	60 (46.5%)	14 (10.8%)	0.35
	Female	27 (20.9%)	11 (8.5%)	12 (9.3%)	4 (3.1%)	
Race, n (%)	White	72 (55.8%)	18 (13.9%)	40 (31%)	14 (10.8%)	0.40
	Black	35 (27.1%)	12 (9.3%)	20 (15.5%)	3 (2.3%)	
	Asian	9 (6.9%)	4 (3.1%)	4 (3.1%)	1 (0.70%)	
	Other	13 (10%)	5 (3.8%)	8 (6.2%)	0 (0%)	
Child-Pugh score, n (%)	Child A	91 (70.5%)	31 (24%)	45 (34.8%)	15 (11.6%)	0.12
	Child B	33 (25.5%)	8 (6.2%)	22 (17%)	3 (2.3%)	
	Child C	5 (3.8%)	0 (0%)	5 (3.8%)	0 (0%)	
Patients’ outcome, n (%)	Alive	71 (55%)	25 (19.3%)	39 (30.2%)	7 (5.4%)	0.12
	Died	52 (40.3%)	12 (9.3%)	29 (22.4%)	11 (8.5%)	
Lobe, n (%) ^¥^	Left	41 (31.7%)	16 (12.4%)	19 (14.7%)	6 (4.6%)	0.26
	Right	84 (65.1%)	23 (17.8%)	49 (37.9%)	12 (9.3%)	
	Both lobes	4 (3.1%)	0 (0%)	4 (3.1%)	0 (0%)	
AFP *, ng/mL		22.5 (6.8–171.93)	9.2 (4–94)	22 (7.8–125)	256.3 (36.3–9621)	0.003
	AFP: alpha fetoprotein * All continuous variables are presented by their median and (interquartile ranges) ^¥^ Total number of lesions is more than the number of patients. More than 1 lesion was identified in 35 patients. ^Δ^ *p* Values of Kruskal–Wallis test reported for continuous variables (age, AFP), and P values of chi square test reported for categorical parameters (sex, race, Child–Pugh score, patients’ outcome, lobe)					

## Data Availability

Data available upon request, please contact Corresponding Author.

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
