# Peer review of "Role of MRI-Derived Radiomics Features in Determining Degree of Tumor Differentiation of Hepatocellular Carcinoma"

_diagnostics, 2022, doi:10.3390/diagnostics12102386_

Round 1
Reviewer 1 Report
Comments
The paper used a radiomics approach to extract features and employed random forest to predict the differentiation grade of HCC. However, there are many related researches at present, and the contribution of this paper is negligible(For example, some studies mentioned below). But most importantly, many conclusions lack realism without relevant experiments.
a) Preoperative prediction of pathological grading of hepatocellular carcinoma using machine learning-based ultrasomics: A multicenter study. https://doi.org/10.1016/j.ejrad.2021.109891
b)Radiomics Analysis of Contrast-Enhanced CT for Hepatocellular Carcinoma Grading. https://doi.org/10.3389/fonc.2021.660509
c) Predicting the grade of hepatocellular carcinoma based on non-contrast-enhanced MRI radiomics signature. https://doi.org/10.1007/s00330-018-5787-2
1. The introduction part lacks citations to related research on the combination of radiomics and artificial intelligence. More recent studies should be cited (e.g. 2020, 2021, 2022).
2. The experimental results are only feature importance and ROC map. There should be enough experiments and analysis to justify the point of the article.
3. Line 201: "...had an out-of-bag error rate of 39.8%, the error decreased to 32.8% (p<0.01) with the addition of the Radiomics features in the multi-class model (Figure 4)." According to the feature importance map, the error rate cannot be obtained. Not enough evidence.
4. Line 218, “…the addition of radiomics… from 27.5% to 22.2% (p<0.01)…from 77.9% to 81.5% (p=0.18).” Where is the source of the quantitative results? More evidence is needed.
5. In discussion: "…we demonstrated that ADC features had better performance in classifying HCC tumors as compared to VE features." No evidence that ADC features had better performance seemed to be found in the experimental section.
6. Likewise, the sentence mentioned below: "From all the features, we identified five that were superior to other features in tumor classification." There are no experiments on the superiority of these five features.
Author Response
Reviewer 1:
Authors aimed to investigate radiomics ability in predicting hepatocellular carcinoma histological degree of differentiation by using volumetric MR imaging parameters. This is an interesting paper. There are several minor concerns.
1) Despite the improved AUC, it might be still not so high. So, it should be discussed as a limitation.
We thank the author for their insightful and helpful comments. We have now added the following sentence to the Limitations as suggested.
“”
2) Conventional radiological features according to HCC histology should be described in the introduction and/or discussion section.
Typical imaging appearance described in Introduction and materials and methods.
Reviewer 2 Report
Authors aimed to investigate radiomics ability in predicting hepatocellular carcinoma histological degree of differentiation by using volumetric MR imaging parameters. This is an interesting paper. There are several minor concerns.
1) Despite the improved AUC, it might be still not so high. So, it should be discussed as a limitation.
2) Conventional radiological features according to HCC histology should be described in the introduction and/or discussion section.
Author Response
Reviewer 2:
The paper used a radiomics approach to extract features and employed random forest to predict the differentiation grade of HCC. However, there are many related researches at present, and the contribution of this paper is negligible(For example, some studies mentioned below). But most importantly, many conclusions lack realism without relevant experiments.
- a) Preoperative prediction of pathological grading of hepatocellular carcinoma using machine learning-based ultrasomics: A multicenter study. https://doi.org/10.1016/j.ejrad.2021.109891
b)Radiomics Analysis of Contrast-Enhanced CT for Hepatocellular Carcinoma Grading. https://doi.org/10.3389/fonc.2021.660509
- c) Predicting the grade of hepatocellular carcinoma based on non-contrast-enhanced MRI radiomics signature. https://doi.org/10.1007/s00330-018-5787-2
- The introduction part lacks citations to related research on the combination of radiomics and artificial intelligence. More recent studies should be cited (e.g. 2020, 2021, 2022).
We thank the author for their constructive comments, we have now cited these articles.
- The experimental results are only feature importance and ROC map. There should be enough experiments and analysis to justify the point of the article.
The aim of our research was two-fold: (1) To investigate radiomics ability in predicting hepatocellular carcinoma histological degree of differentiation by using volumetric MR imaging parameters – the comparison of ROC curves serves this purpose, and (2) To identify the radiomics features that were the best at differentiation – the feature importance helps identify the same. This is typical of what prior researchers have done, including the three articles cited above. Please also note that while the above-referenced MRI-radiomics article only used T1 and T2 weighted images, we used radiomics features of ADC maps and enhancement maps. We have added a paragraph in Introduction to address the motivation and prior work in nonradiomics data as well.
- Line 201: "...had an out-of-bag error rate of 39.8%, the error decreased to 32.8% (p<0.01) with the addition of the Radiomics features in the multi-class model (Figure 4)." According to the feature importance map, the error rate cannot be obtained. Not enough evidence.
The feature map provides feature importance and is not an indicator of the error rates. The error rates were extracted from the random forest classifier algorithm as previously explained in the methods. We thank the reviewer for pointing out the awkward wording, and have now modified the sentence, and moved this portion (referencing Figure 4) to the prior section.
- Line 218, “…the addition of radiomics… from 27.5% to 22.2% (p<0.01)…from 77.9% to 81.5% (p=0.18).” Where is the source of the quantitative results? More evidence is needed.
We have now added Figure 6 showing the ROC curves for the two-class random forest classifier model with and without radiomics features as well.
- In discussion: "…we demonstrated that ADC features had better performance in classifying HCC tumors as compared to VE features." No evidence that ADC features had better performance seemed to be found in the experimental section.
We have modified this sentence to reflect the results more appropriately as follows – “we demonstrated that ADC radiomics features were among the top classifiers in variable importance ranking for classifying HCC tumors as compared to VE features.”
- Likewise, the sentence mentioned below: "From all the features, we identified five that were superior to other features in tumor classification." There are no experiments on the superiority of these five features.
The superiority is based on variable ranking as obtained from the minimum depth of the maximal subtree. The methods section describes how these are obtained.
Round 2
Reviewer 1 Report
Line:200,"...with (top) and without (bottom) radiomics-derived...",is there an error in the location of the Figure?
Author Response
We thank the reviewer for bringing this to our attention. We changed the figure legend accordingly.